# Curriculum Offline Imitating Learning

**Minghuan Liu**[1]* **Hanye Zhao**[1]* **Zhengyu Yang**[1] **Jian Shen**[1]
**Weinan Zhang**[1]† **Li Zhao**[2] **Tie-Yan Liu**[2]
[1] Shanghai Jiao Tong University, [2] Microsoft Research
{minghuanliu, fineartz, zyyang, rockyshen, wnzhang}@sjtu.edu.cn,
{lizo,tyliu}@microsoft.com

## Abstract

Offline reinforcement learning (RL) tasks require the agent to learn from a pre-collected dataset with no further interactions with the environment. Despite the potential to surpass the behavioral policies, RL-based methods are generally impractical due to the training instability and bootstrapping the extrapolation errors, which always require careful hyperparameter tuning via online evaluation. In contrast, offline imitation learning (IL) has no such issues since it learns the policy directly without estimating the value function by bootstrapping. However, IL is usually limited in the capability of the behavioral policy and tends to learn a mediocre behavior from the dataset collected by the mixture of policies. In this paper, we aim to take advantage of IL but mitigate such a drawback. Observing that behavior cloning is able to imitate neighboring policies with less data, we propose *Curriculum Offline Imitation Learning (COIL)*, which utilizes an experience picking strategy for imitating from adaptive neighboring policies with a higher return, and improves the current policy along curriculum stages. On continuous control benchmarks, we compare COIL against both imitation-based and RL-based methods, showing that it not only avoids just learning a mediocre behavior on mixed datasets but is also even competitive with state-of-the-art offline RL methods.

## 1 Introduction

Offline reinforcement learning (RL), or batch RL, aims to learn a well-behaved policy from arbitrary datasets without interacting with the environment. This setting is generally a more practical paradigm than online RL since it is expensive or dangerous to interact with the environment in most real-world applications. Typically, two main kinds of offline datasets are considered in previous offline RL works [5, 18]: one contains transitions sampled by a single behavioral policy; the other includes a buffer collected by a mixture of policies.

Two main approaches have been deeply investigated for Offline RL. First, RL-based methods, in particular, Q-learning and policy gradient-based algorithms [13, 7, 12], have the potential to outperform the behavioral policy. However, they always suffer from serious bootstrapping errors and training instability. This shortcoming makes such algorithms impractical to be utilized since too many hyperparameters need to be tuned to achieve a good performance, and it is hard to evaluate a suitable model in an offline manner, as revealed in [18]. In contrast, offline imitation learning [1, 17, 2], specifically, behavior cloning (BC), can always stably learn to perform as the behavioral policy, which may be helpful under single-behavior datasets. However, BC may fail in learning a good behavior under a diverse dataset containing a mixture of policies (both goods and bads).

**Quantity-quality dilemma on mixed dataset.** As a supervised learning technique, BC is not easy to fulfill a desired result, especially on a mixed dataset. Specifically, it requires both quantity and quality

---

*Equal contribution. †Corresponding author. Codes are available at `https://github.com/apexrl/COIL`.

35th Conference on Neural Information Processing Systems (NeurIPS 2021).

of the demonstration data, which can hardly be satisfied in offline RL tasks. Directly mimicking the policy from a mixed dataset that contains bad-to-good demonstrations can be regarded as imitation learning from a mediocre behavior policy. To achieve the best performance of the dataset, a naive idea is to imitate the top trajectories ordered by its return. However, such a simple strategy will reach the *quantity-quality dilemma* on the mixed dataset. For example, Fig. 1a illustrates the ordered trajectories on the `Walker-final-dataset`, which contains the whole training experience sampled by an online training agent. Different BC agents are trained by the top 10%, 25%, 50%, and 100% trajectories, but none of them gets rid of a mediocre performance, as shown in Fig. 1b. Typically, on such dataset, less data owns higher quality but less quantity, and thus cause serious compounding error problems [21, 10, 6]; on the other hand, more data provides a larger quantity, yet its mean quality becomes worse. In this work, we aim to solve such a dilemma and exploit the most potential of IL to derive a stable and practical algorithm reaching the best performance of a given dataset.

Our intuition comes from the observation that under RL scenarios, the agent can imitate a neighboring policy with much fewer samples. This observation promotes a curriculum solution for the above challenge. Specifically, for mixed datasets, the agent can adaptively imitate the better neighboring policies step by step and finally reach the optimal behavior policy of the dataset.

**Our work.** We propose Curriculum Offline Imitation Learning (COIL), a simple yet effective offline imitation learning framework for offline RL. At each training iteration, COIL improves the current policy with the data sampled by neighboring policies. To achieve that, COIL utilizes an adaptive experience picking strategy and a return filter to select proper trajectories from the offline dataset for the current level of the agent and thus produces stages of the curriculum. Notably, COIL stops with a close-to-data-optimal policy without finding the best model under online evaluation.

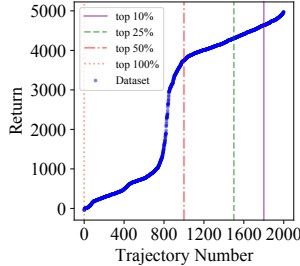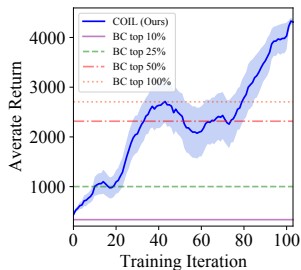

(a) Ordered trajectories.    (b) Returns of BC v.s. COIL.

Figure 1: Examples of the quality-quantity dilemma for BC. (a) Trajectories of the Walker2d-final dataset ordered by their accumulated return. (b) Performances of behavior cloning (BC) for learning the top 10%, 25%, 50%, and 100% trajectories of the dataset.

This feature allows to deploy the algorithm in practical problems. In experiments, we show the effectiveness of COIL on various kinds of offline datasets. Fig. 1 offers a quick review of our results: depending solely on BC, COIL can learn from scratch to reach the best performance of the given dataset.

**Contributions.** To summarize, the main technical contributions of this paper are as follows.

- We highlight how the discrepancy between the target policy and the initialized policy affects the number of samples required by BC (Section 3);
- Depending on BC, we propose a practical and effective offline RL algorithm with a practical neighboring experience picking strategy that ends with a good policy (Section 4);
- We present promising comparison results with comprehensive analysis for our algorithm, which is competitive to the state-of-the-art methods (Section 6).

## 2 Preliminaries

**Notations.** We consider a standard Markov Decision Process (MDP) as a tuple $\mathcal{M} = \langle \mathcal{S}, \mathcal{A}, \mathcal{T}, \rho_0, r, \gamma \rangle$, where $\mathcal{S}$ is the state space, $\mathcal{A}$ represents the action space, $\mathcal{T} : \mathcal{S} \times \mathcal{A} \times \mathcal{S} \to [0, 1]$ is the state transition probability distribution, $\rho_0 : \mathcal{S} \to [0, 1]$ is the initial state distribution, $\gamma \in [0, 1]$ is the discounted factor, and $r : \mathcal{S} \times \mathcal{A} \to \mathbb{R}$ is the reward function. The goal of RL is to find a policy $\pi(a|s) : \mathcal{S} \times \mathcal{A} \to [0, 1]$ that maximizes the expected cumulative discounted rewards (or called return) along a trajectory $\tau$: $R(\tau) = \sum_{t=0}^{T} \gamma^t r_t$. The dataset $\mathcal{D}$ consists of trajectories $\{\tau_1^N\}$ that are the collected by a mixture of bad-to-good policies, where a trajectory $\tau_i = \{(s_0^i, a_0^i, s_0'^i, r_0^i), (s_1^i, a_1^i, s_1'^i, r_1^i), \cdots, (s_{h_i}^i, a_{h_i}^i, s_{h_i}'^i, r_{h_i}^i)\}$, and $h_i$ is the horizon of $\tau_i$. For any dataset, we assume a behavior policy $\pi_b$ that collects such data and its empirical estimation $\hat{\pi}_b$ can be induced from $\mathcal{D}$ as $\hat{\pi}_b(a|s) = \frac{\sum_{(s', a') \in \mathcal{D}} \mathbb{I}[s'=s, a'=a]}{\sum_{s' \in \mathcal{D}} \mathbb{I}[s'=s]}$, where $\mathbb{I}$ is the indicator function. We further introduce a common used term, occupancy measure, which is

defined as the discounted occurrence probability of states or state-action pairs under policy $\pi$: $\rho_\pi(s,a) = \sum_{t=0}^\infty \gamma^t P(s_t = s, a_t = a|\pi) = \pi(a|s) \sum_{t=0}^\infty \gamma^t P(s_t = s|\pi) = \pi(a|s)\rho_\pi(s)$. With such a definition we can write down that $\mathbb{E}_\pi[\cdot] = \sum_{s,a} \rho_\pi(s,a)[\cdot] = \mathbb{E}_{(s,a)\sim\rho_\pi}[\cdot]$.

**Definition 1.** *The partial order of different policies is defined as the relative return quantity that a policy can achieve when deployed in the environment. Formally, given two policies $\pi_1$ and $\pi_2$:*

$$\pi_1 \preceq \pi_2 \Leftrightarrow R_1 \preceq R_2 \tag{1}$$

Therefore, by definition, in a mixed dataset $\mathcal{D}$ that is collected by $K$ different policies $\pi_1, \cdots, \pi_K$, the optimal behavior policy $\pi^*$ can be determined such that for $\forall i \in [1, K], \pi_i \preceq \pi^*$.

**Curriculum learning.** Curriculum learning design and construct a *curriculum* automatically as a sequence of tasks $G_1, \ldots, G_N$ to train on, such that the efficiency or performance on a target task $G_t$ is improved. The expected loss on the $j^{\text{th}}$ task is denoted $\mathcal{L}_j$.

# 3 Empirical Observations and Theoretic Analysis

In this section, we begin with empirical observations that motivate the core idea of our method, followed by the theoretical analysis to support our motivation. Generally, we aim to investigate how the asymptotic performance of BC is affected by the discrepancy between the demonstrated policy and the initialized imitating policy. Previous research shows that BC tends to fail with a small number of high-quality demonstrations but can learn well from large-quantity and high-quality data [10, 8]. On the contrary, we find that the requirement of quantity can be highly relaxed as the similarities between the demonstrated policy and the initialized imitating policy increase.

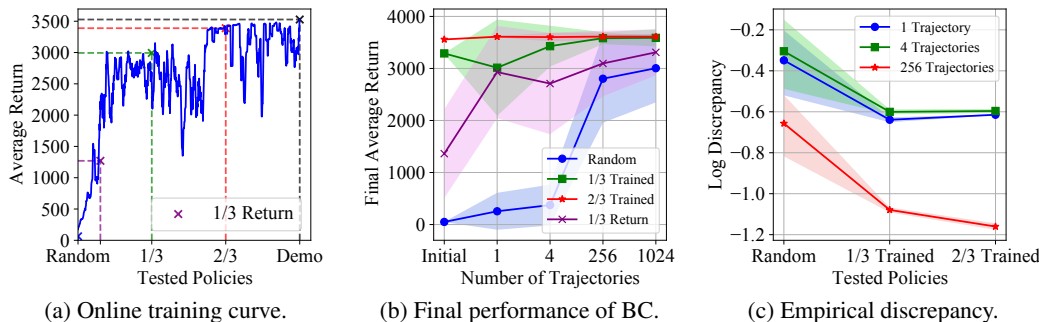

(a) Online training curve.    (b) Final performance of BC.    (c) Empirical discrepancy.

Figure 2: (a) Online training curves of an SAC agent trained on the Hopper environment, where the crosses and dashed lines indicate the stage of selected policies. (b) Final performances achieved by imitating the demo policy using BC, initialized with different stages of policies. The curves depict the fact that close-to-demonstration policy can easily imitate the demonstrated policy with fewer samples. (c) Empirical estimation on the discrepancy between the initialized policy and the trained policy outside the support of the demonstrations. Initialized with a closer-to-demo policy always enjoys more minor discrepancy.

## 3.1 BC with Different Initialization

To construct the motivating example, we choose Hopper as the testbed, and train an SAC agent until convergence to sample various counts of trajectories as the demonstration data. We then take the online-trained policy checkpoints at different training iterations as the initiated policy to train an IL agent. Particularly, the first agent adopts the `Random` policy to imitate the demonstration by BC; the second uses the policy of `1/3 Return` and the other two agents start with `1/3 Trained` and `2/3 Trained` policy separately, in terms of training iterations (see Fig. 2a).

The results are shown in Fig. 2b, where we illustrate the average return of each agent given the different number of demonstrations (the exact quantitative results can be found in Appendix E.1). The results show that initialized with a `Random` policy, the agent can only learn to imitate the demonstrated policy well with a large number of samples[2]; in addition, both `1/3 Return` and `1/3 Trained` policies can achieve a sub-optimal performance with fewer samples, where the `1/3`

---

[2] We note that the `Random` agent can only work well on large datasets with normalized state space; however the other agents learn well upon raw states.

`Trained` one is more efficient. In comparison, the `2/3 Trained` agent, which is the closest to the optimal policy, can stably achieve the best performance with all amounts of trajectories.

## 3.2 Theoretical Analysis

Beyond these observations, we are inquisitive to find a theoretical explanation to support our claim. Standing on the primary result of existing works, we obtain a performance bound of BC with a possible solution to handle the quantity-quality dilemma.

**Theorem 1** (Performance bound of BC). *Let $\Pi$ be the set of all deterministic policy and $|\Pi| = |A|^{|S|}$. Assume that there does not exist a policy $\pi \in \Pi$ such that $\pi(s^i) = a^i, \forall i \in \{1, \cdots, |\mathcal{D}|\}$. Let $\hat{\pi}_b$ be the empirical behavior policy as well as the corresponding state marginal occupancy is $\rho_{\hat{\pi}_b}$. Suppose BC begins from initial policy $\pi_0$, and define $\rho_{\pi_0}$ similarly. Then, for any $\delta > 0$, with probability at least $1 - \delta$, the following inequality holds:*

$$D_{\mathrm{TV}}(\rho_\pi(s,a)\|\rho_{\pi_b}(s,a)) \le c(\pi_0, \pi_b, |\mathcal{D}|)$$

$$where \quad c(\pi_0, \pi_b, |\mathcal{D}|) = \frac{1}{2}\sum_{s \notin \mathcal{D}} \rho_{\pi_b}(s) + \frac{1}{2}\sum_{s \notin \mathcal{D}} |\rho_\pi(s) - \rho_{\pi_0}(s)| + \underbrace{\frac{1}{2}\sum_{s \notin \mathcal{D}} |\rho_{\pi_0}(s) - \rho_{\pi_b}(s)|}_{\textit{initialization gap}}$$

$$+ \underbrace{\frac{1}{2}\sum_{s \in \mathcal{D}} |\rho_\pi(s) - \rho_{\hat{\pi}_b}(s)| + \frac{1}{|\mathcal{D}|}\sum_{i=1}^{|\mathcal{D}|} \mathbb{I}\left[\pi(s^i) \ne a^i\right]}_{\textit{BC gap}} + \underbrace{\left[\frac{\log|\mathcal{S}| + \log(2/\delta)}{2|\mathcal{D}|}\right]^{\frac{1}{2}} + \left[\frac{\log|\Pi| + \log(2/\delta)}{2|\mathcal{D}|}\right]^{\frac{1}{2}}}_{\textit{data gap}}$$

$$(2)$$

The proof can be found in Appendix B.1. Theorem 1 shows the upper bound of the state-action distribution between the imitating policy $\pi$ and the behavior policy $\pi_b$, which consists of three important terms: the *initialization gap*, the *BC gap* and the *data gap*. Specifically, the *BC gap* arises from the empirical error and the difference between the imitating policy and the empirical behavior policy, which is corresponding to the training procedure. The *data gap*, however, depends on the number of samples and complexity of the state space, acting as an intrinsic gap due to the dataset and the environment. As for the *initialization gap*, it is in the form of distance between the state marginal distribution of the initial policy $\pi_0$ and behavior policy $\pi_b$ out of the dataset. Notice that the second term in Eq. (2) relates to the distance between the state marginal of the initial policy $\pi_0$ and the learned policy $\pi_b$ outside the data support, which is hard to measure theoretically due to the Markov property of the environment dynamics. Therefore, we estimate the empirical discrepancy outside the dataset $\frac{1}{2}\sum_{s \notin \mathcal{D}} |\hat{\rho}_\pi(s) - \hat{\rho}_{\pi_0}(s)|$ for this term[3]. The results shown in Fig. 2c, as expected, suggests that the second term in Eq. (2) in fact decreases as the initialized policy gets close to the demonstrated policy because of the poor generalization on unseen states, and the error can be further reduced with a larger dataset.

Such analysis brings a possible theoretical explanation to our empirical intuition Section 3.1. Generally, given the same discrepancy $c(\pi_0, \pi_b, |\mathcal{D}|) = C$, if the initialized policy narrows down the *initialization gap* as is close to the demonstrated policy, then the requirement for more samples to minimize the *data gap* can be relaxed. This may seem unreasonable in the learning theory in the traditional supervised learning domain. However, under the RL scenario, the performance of a policy depends on the accumulated reward along the rollout trajectories, which will lead to serious compounding error problems [21, 20]. Therefore, as the distance between the initialized policy and the demonstrated policy gets closer, the generalization errors of the learned policy can be reduced.

**Brief conclusion.** Both the experimental and the theoretical results indicate an interesting fact that the asymptotic performance of BC is highly related to the discrepancy between the initialized policy and the demonstrated policy. Specifically, a close-to-demonstration policy can easily imitate the demonstrated policy with fewer samples. On the contrary, when the distance between the initialized policy and the demonstrated policy is far, then successfully mimicking the policy will require much more samples. Such an observation motivates the intuition for proposing our Curriculum Offline Imitation Learning (COIL) in the following literature. The key insight enabling COIL is adaptively imitating the close policies with a small number of samples and finally terminates with the optimal behavior policy of the dataset.

---

[3]For implementation details, see Appendix C.1.

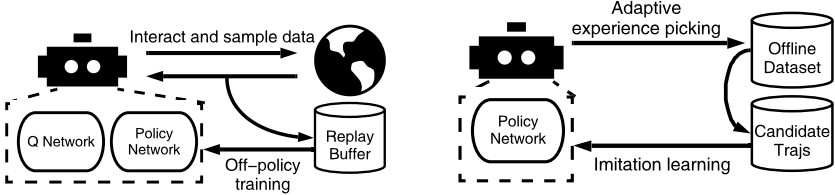

(a) Online off-policy training.   (b) Curriculum offline imitation learning.

Figure 3: Comparison between online off-policy training and curriculum offline imitation learning.

## 4 Curriculum Offline Imitation Learning

### 4.1 Online RL as Imitating Optimal Policies

Before starting to formulate our methodology, we first introduce the formulation of the online RL training. Typically, if we treat an optimization step of the policy as a training stage, an RL online learning algorithm can be realized as on-policy / off-policy depending on whether the agent is trained using the data collected by policies in the previous training stage. Taking off-policy RL as an example: beginning with a randomized policy $\pi^0$, at every training stage $i$, the agent uses its policy $\pi^i$ to interact with the environment to collect trajectory $\tau^i$ and save it to replay buffer $\mathcal{B}$. The agent then samples several state-action pairs from $\mathcal{B}$ and take an optimization step to get policy $\pi^{i+1}$, towards obtaining the most accumulated rewards:

$$\underset{\pi}{\text{maximize}}\ \mathbb{E}_{\tau \sim \pi}[R(\tau)] .\tag{3}$$

Under the principle of maximum entropy, we can model the distribution of trajectories sampled by the optimal policy as a Boltzmann distribution [27, 4] as:

$$P^*(\tau) \propto \exp\left(R(\tau)\right)\tag{4}$$

With such a model, trajectories with higher rewards are exponentially more preferred. And finding the optimal policy through RL is equivalent to imitating the optimal policy modeled by Eq. (4) [4, 3]:

$$\underset{\pi}{\text{minimize}}\ D_{KL}(P_\pi(\tau)\|P^*(\tau)) ,\tag{5}$$

where $P_\pi(\tau) = \rho(s_0)\sum_{t=0}^{T} \mathcal{T}(s_{t+1}|s_t, a_t)\pi(a_t|s_t)$ is the distribution of generating a trajectory $\tau$ according to policy $\pi$. Thus, $\pi^{i+1}$ is updated follows the direction of minimizing the KL divergence:

$$\pi^{i+1} = \pi^i - \nabla_\pi D_{KL}(P_\pi(\tau)\|P^*(\tau))\tag{6}$$

### 4.2 Offline RL as Adaptive Imitation

Compared with online policy training, the offline agent can only have a pre-collected dataset for policy training. Such dataset could be generated by a single policy, or collected by kinds of policies. In analogy to online RL, a similar solution on offline tasks can be imitating the optimal behavior policy from the dataset in an offline way. However, as we show before, offline IL methods, like BC, are only capable of matching the performance of the behavior policy, but hard to reach a good performance on the mixed dataset due to the quantity-quality dilemma.

#### 4.2.1 Leverage Behavior Cloning with Curriculum Learning

In Section 3 we have seen evidence that a possible solution to the quantity-quality dilemma could be adaptive imitation through the offline dataset. An overview of such an adaptive imitation learning diagram compared with online RL is shown in Fig. 3. Formally, with dataset $\mathcal{D} = \{\tau\}_1^N$, at every training stage $i$, the agent updates its policy $\pi^i$ by adaptively selecting $\tau \sim \tilde{\pi}^i$ from $\mathcal{D}$ as the imitating target such that:

$$\begin{aligned} \pi^{i+1} =\ &\pi^i - \nabla_\pi D_{KL}(P_{\tilde{\pi}^i}(\tau)\|P_\pi(\tau)) \\ &\text{s.t. } \mathbb{E}_{\tilde{\pi}}\left[D_{KL}(\tilde{\pi}^i(\cdot|s)\|\pi^i(\cdot|s))\right] \leq \epsilon \\ &\quad R_{\tilde{\pi}}^i - R_\pi^i \geq \delta \end{aligned}\tag{7}$$

where $\epsilon$ and $\delta$ are small positive numbers that limit the difference between the demonstrated policy $\tilde{\pi}^i$ and the learner $\pi$, and prevent $\pi$ from learning poorly behaved policies. Correspondingly, each training

iteration $i$ creates a curriculum automatically such that a task $G_i$ is to imitating the closet demonstrated policy $\tilde{\pi}_i$ with $\mathcal{L}_i^{\pi} = D_{KL}(P_{\tilde{\pi}_i}(\tau) \| P_{\pi}(\tau))$, and the target task $G_t$ is to imitate the optimal policy $\tilde{\pi}^*$. Specifically, we aims to construct a finite sequence $\pi^0, \tilde{\pi}^1, \pi^1, \tilde{\pi}^2, \pi^2, \cdots, \tilde{\pi}^N, \pi^N$ such that $\pi^i \preceq \pi^{i+1}$, where $\tilde{\pi}^i$ is characterized by its trajectory, and is picked from $\mathcal{D}$ based on the current policy $\pi^{i-1}$; $\pi^i$ is the imitation result taken $\tilde{\pi}^i$ as the target policy.

In the following sections, we explain how our algorithm is designed to achieve Eq. (7) that leads the target policy $\tilde{\pi}$ to finally collapse into the optimal behavior policy $\tilde{\pi}^*$, while solving the quality-quantity dilemma of BC to avoid getting a mediocre result.

### 4.2.2 Adaptive Experience Picking by Neighboring Policy Assessment

We first provide a practical solution to evaluate whether $\mathbb{E}_{\tilde{\pi}}[D_{KL}(\tilde{\pi}(\cdot|s) \| \pi(\cdot|s))] \leq \epsilon$. In other words, we want to know whether a trajectory $\tau_i \in \mathcal{D}$ is sampled by a neighboring policy. This can be regarded as finding a policy whose importance sampling ratio is near to 1, and thus a lot of density ratio estimation works can be referred to [16, 26]. However, such estimation requires extra costs on training neural networks, and the estimation is inaccurate with fewer data points. Therefore, in this paper, we design a simple yet efficient neighboring policy assessment principle instead that brings the algorithm into practice.

We assume that each trajectory is sampled by a single policy. Beyond such a slight and practical assumption, let $\pi$ be the current policy and trajectory $\tau_{\tilde{\pi}} = \{(s_0, a_0, s_0', r_0), \cdots, (s_h, a_h, s_h', r_h)\}$ is collected by an unknown deterministic behavior policy $\tilde{\pi}$ with exploration noise such that $\mathbb{E}_{(s,a) \in \tau_{\tilde{\pi}}}[\log \tilde{\pi}(a_t|s_t)] \geq \log(1 - \beta)$, where $\beta$ denote the portion of exploration. In this way, we find a practical solution that relaxes the KL-divergence constraint through an observation:

**Observation 1.** *Under the assumption that each trajectory $\tau_{\tilde{\pi}}$ in the dataset $\mathcal{D}$ is collected by an unknown deterministic behavior policy $\tilde{\pi}$ with an exploration ratio $\beta$. The requirement of the KL divergence constraint $\mathbb{E}_{\tilde{\pi}}[D_{KL}(\tilde{\pi}(\cdot|s) \| \pi(\cdot|s))] \leq \epsilon$ suffices to finding a trajectory that at least $1 - \beta$ state-action pairs are sampled by the current policy $\pi$ with a probability of more than $\epsilon_c$ such that $\epsilon_c \geq 1/\exp \epsilon$, i.e.:*

$$\mathbb{E}_{(s,a) \in \tau_{\tilde{\pi}}}[\mathbb{I}(\pi(a|s) \geq \epsilon_c)] \geq 1 - \beta , \tag{8}$$

The corresponding deviation is shown in Appendix B.2. Therefore, to find whether $\tau_{\tilde{\pi}}$ is sampled by a neighboring policy, we calculate the probability of sampling $a_t$ at state $s_t$ by $\pi$ in $\tau$ for every timestep $\tau_{\tilde{\pi}}(\pi) = \{\pi(a_0|s_0), \cdots, \pi(a_h|s_h)\}$, where $h$ is the horizon of the trajectory. In practice, instead of fine-tuning $\epsilon$ and $\beta$, we heuristically set $\beta = 0.05$ as an intuitive ratio of exploration. As for $\epsilon_c$, we let the agent choose the value through finding $N$ nearest policies that matches Eq. (8).

### 4.2.3 Return Filter

We now present how to ensure the second constraint $R_{\tilde{\pi}} - R_{\pi} \geq \delta$, which is designed to refrain the performance from getting worse by imitating to a poorer target than the current level of the imitating policy. In a practical offline scenario, it is impossible to get the accurate return of the current policy, but we can evaluate its performance based on the current curriculum. To this end, we adopt a return filtering mechanism that filtrates the useless, poor-behaved trajectories.

In practice, we initialize the return filter $V$ with 0, and update the value at each curriculum. Specifically, if we choose $\{\tau\}_1^n$ from $\mathcal{D}$ at iteration $k$, then $V$ is updated by moving average:

$$V_k = (1 - \alpha) \cdot V_{k-1} + \alpha \cdot \min\{R(\tau)\}_1^n \tag{9}$$

where $\{R(\tau)\}_1^n$ is the accumulated reward set of trajectories $\{\tau\}_1^n$, and $\alpha$ is the moving window determining the filtering rate. Then, the dataset is updated as $\mathcal{D} = \{\tau \in \mathcal{D} \mid R(\tau) \geq V\}$.

### 4.2.4 Overall Algorithm

Combining the adaptive experience picking strategy and the return filter, we finally get the simple and practical curriculum offline imitation learning (COIL) algorithm. To be specific, COIL holds an experience pool that contains the candidate trajectories to be selected. Every training time creates a stage of the curriculum where the agent selects appropriate trajectories as the imitation target from the pool and learns them via direct BC. After training, the used experience will be cleaned from the pool, and the return filter also filtrates a set of trajectories. An attractive property of COIL is that it

has a terminating condition that stops the algorithm automatically with a good policy when there is no candidate trajectory to be selected. This makes it easier to be applied in real-world applications without further finding the best-learned policy checkpoints under online evaluation as the previous algorithms do. The step-by-step algorithm is shown in Algo. 1.

It is worth noting that COIL has only two critical hyperparameters, namely, the number of selected trajectories $N$ and the moving window of the return filter $\alpha$, both of which can be determined by the property of the dataset. Specifically, $N$ is related to the average discrepancy between the sampling policies in the dataset; $\alpha$ is influenced by the changes of the return of the trajectories contained in the dataset. In the ablation study Section 6.3 and Appendix E.2, we demonstrate how we select different hyperparameters for different datasets.

## 5 Related Work

As a long-studied branch of RL, numerous solutions have been developed for offline RL with both model-free [7, 12] and model-based [25, 11] algorithms. We briefly discuss the former from two categories, including RL-based methods and imitation-based methods.

**RL-based methods.** A naive thought is to apply off-policy RL algorithms such as SAC [9] and DDPG [14] directly. However, as previous researchers [7, 12] reveal, those online algorithms fail to work due to the severe extrapolation error or out-of-distribution problem. Thence, kinds of specifically designed algorithms have been proposed for offline RL. For instance, BCQ [7] adopts a perturbation model to disturb the action sampled by a BC module to conduct Q-learning on the offline data. BEAR [12] augments constraints on the policy to avoid out-of-distribution actions, over which it maximizes the approximate Q function. The current state-of-the-art algorithm CQL [13] performs strict constraints on the Q-function to learn an expectation lower bound of the true value, avoiding overestimation on out-of-distribution data. Typical drawbacks for these RL-based algorithms are serious bootstrapping errors and training instability that requires careful hyperparameter tuning with online evaluation. In comparison, imitation-based methods offer a candidate to mimic the demonstration.

**Imitation-based methods.** Another inspiration grows from the technique of offline imitation learning on how to learn from the demonstration. Most of these methods take the idea of behavior cloning (BC) [1] that utilizes supervised learning to learn to act. However, due to the quantity-quality dilemma, BC limits mediocre performance on many datasets with mixed samples. Therefore, BAIL [2] constructs the upper-envelope on the value of the data and selects the best actions to imitate at each state and learn the policy based on BC. The main problem in BAIL is the requirement of regressing the value function with tricks to compute a value in the infinite horizon and the hyperparameter threshold on determining the best action. ABM [22], MARWIL [23] and AWR [17] all take the idea of using an exponentially weighted version of BC, where the weights are determined by different forms of advantage function. These methods also require estimating the value function based on the offline data, which can be unstable and need many hyperparameters to control the learning procedure. Besides, it is also hard to tune an appropriate scale for the advantage for imitating the behavior policy. Compared with these methods, our COIL only has few important hyperparameters to be tuned without regressing any value function to achieve a stable performance.

## 6 Experiments

Alongside the simple algorithm, we surprisingly find that COIL not only alleviates the quantity-quality dilemma but also achieves efficient and stable performance against competitive offline methods. Furthermore, we carry out a comprehensive analysis of the algorithm behind the phenomenon.

### 6.1 Offline Learning from Online Learning Experience

We are curious about the performance of COIL on the *final buffer* datasets containing a complete training experience of an online agent [7], because such dataset is mixed with various levels of policies. Therefore direct BC will easily fail as shown in Section 1. In our case, we first train an SAC agent from scratch to convergence and save all the training experience from its interactions with the environment, and therefore including exploration actions. Specifically, we conduct our experiments

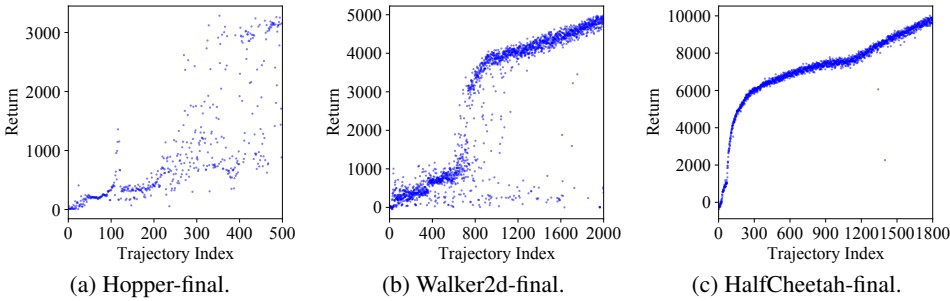

Figure 4: Trajectories in final datasets, sorted in the online training order.

on three continuous control tasks: Hopper, Walker2d, and HalfCheetah. Fig. 4 illustrates trajectories in each dataset, sorted in the collected order.

To show the effectiveness of COIL, we compare it with several strong baselines, including the state-of-the-art offline RL algorithm CQL [13], and imitation-based methods AWR [17] and BAIL [2]. For CQL, AWR and BAIL we use their open-source implementation with their default hyperparameter, and for BC we test the same implementation as COIL. The quantitative converged results are listed in Tab. 1, from which we observe that COIL substantially outperforms the other baselines for the *final buffer* dataset. Also, BAIL and AWR can not always find the optimal behavior due to the difficulty of its hyperparameters tuning and value regression. Specifically, compared with BC that learns a mediocre policy, COIL reaches the performance close to the optimal policy.

To further illustrate how our algorithm works, we also record the oracle online order of the trajectory sampled by the offline agent, as shown in Fig. 5. Notably, COIL keeps a similar training path as the online agent, thanks to the experience picking strategy and the return filter. The corresponding learning curves shown in Fig. 5 are stable and following similar shapes as the ordered datasets. Notably, COIL finally terminates with a near-data-optimal policy, suggesting a nice property that the last offline model can be a great model for deployment, unlike previous algorithms that relies on online evaluation to select the best checkpoint.

Table 1: Average performances on *final* datasets, the means and standard deviations are calculated over 5 random seeds. *Behavior* shows the average performance of the behavior policy that collects the data.

| Dataset | Expert (SAC) | Behavior | BC | AWR | BAIL | CQL | COIL (Ours) |
|---|---|---|---|---|---|---|---|
| hopper-final | 3163.3 (44.4) | 974.5 | 1480.4 (800.2) | 1609.7 (489.7) | 2296.9 (915.9) | 501.5 (227.5) | **2872.5 (133.9)** |
| walker2d-final | 4866.03 (68.6) | 2684.9 | 2099.6 (2101.3) | 3213.8 (1682.9) | **4236.2 (1531.1)** | 2604.3 (1937.6) | **4391.3 (697.8)** |
| halfcheetah-final | 9739.1 (113.6) | 7122.4 | 6125.6 (3910.9) | 7600.9 (1153.4) | 9745.0 (880.3) | **10882.0 (1042.7)** | 9328.5 (1940.6) |

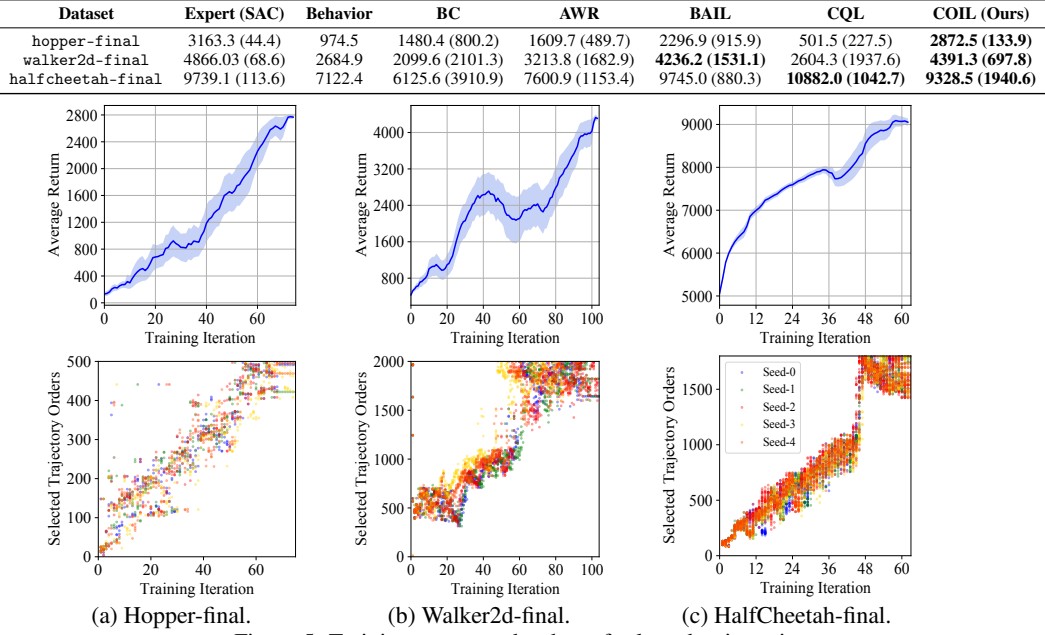

Figure 5: Training curves and orders of selected trajectories.

## 6.2 D4RL Benchmark

To further show the power of COIL, we conduct comparison experiments on a common-used D4RL benchmark [5] in Tab. 2. For BC we include the BC results reported in Fu et al. [5] (denoted as *BC (D4RL)*) and our implementation of BC (denoted as *BC (Ours)*). Besides, we compare BAIL [2],

MOPO [25], and the state-of-the-art results reported in Fu et al. [5] (denoted as *SoTA*). As our expected, BC is able to approach or outperform the performance of the behavior policy on the datasets generated from a single policy, marked as "`random`", and "`medium`", but still remains a gap between the optimal behavior policy (*Best* $1\%$ column). As a comparison, COIL achieves the performance of the optimal behavior policy on most datasets. As is noticed that doing so will allow COIL to beat or compete with the state-of-the-art results. To our surprise, on the `halfcheetah` domain, previous RL-based offline algorithms have the potential to surpass the optimal behavior policy (although with careful hyperparameter tuning), showing the advantage of RL-based learning. It is worth noting that for model-based algorithm like MOPO, it behaves well on the `medium-replay` datasets due to the sufficient data to learn a good environment model; but it can hardly outperform SoTA model-free results on other datasets.

Table 2: Average performance on D4RL datasets. Results in gray columns is our implementation that are tested among 5 random seeds. The other results are based on numbers reported in D4RL among 3 random seeds without standard deviations. *Best 1%* shows the average return of the top 1% best trajectories, representing the performance of the optimal behavior policy; *Behavior* shows the average performance of the dataset.

| Dataset | Expert (D4RL) | Behavior | Best 1% | BC (D4RL) | BC (Ours) | COIL (Ours) | BAIL | MOPO | SoTA (D4RL) |
|---|---|---|---|---|---|---|---|---|---|
| hopper-random | 3234.3 | 295.1 | 340.4 | 299.4 | 330.1 (3.5) | 378.5 (15.2) | 318.0 (5.1) | **432.6** | 376.3 |
| hopper-medium | 3234.3 | 1018.1 | 3076.4 | 923.5 | 1690.1 (852.0) | **3012.0 (332.2)** | 1571.5 (900.7) | 862.1 | 2557.3 |
| hopper-medium-replay | 3234.3 | 466.9 | 1224.8 | 364.4 | 853.6 (397.5) | 1333.7 (271.1) | 808.7 (192.5) | **3009.6** | 1227.3 |
| hopper-medium-expert | 3234.3 | 1846.8 | 3735.7 | **3621.2** | 3527.4 (504.1) | 3615.5 (168.9) | 2435.9 (1265.2) | 1682.0 | 3588.5 |
| walker2d-random | 4592.3 | 1.1 | 25.0 | 73.0 | 171.0 (59.3) | 320.5 (70.7) | 130.8 (87.2) | **597.1** | 336.3 |
| walker2d-medium | 4592.3 | 496.4 | 3616.8 | 304.8 | 1521.9 (1381.3) | 2184.5 (1279.2) | 1242.4 (1545.7) | 643.0 | **3725.8** |
| walker2d-medium-replay | 4592.3 | 356.6 | 1593.7 | 518.6 | 715.0 (406.5) | 1439.9 (347.0) | 532.9 (359.0) | **1961.1** | 1227.3 |
| walker2d-medium-expert | 4592.3 | 1059.7 | 5133.4 | 297.0 | 3488.6 (1815.1) | 4012.3 (1463.0) | 3633.9 (1839.7) | 2526.0 | **5097.3** |
| halfcheetah-random | 12135.0 | -302.6 | -85.4 | -17.9 | -124.3 (60.6) | -0.3 (0.7) | -96.4 (49.7) | 3957.2 | **4114.8** |
| halfcheetah-medium | 12135.0 | 3944.9 | 4327.7 | 4196.4 | 3276.4 (1500.7) | 4319.6 (243.7) | 4277.6 (564.9) | 4987.5 | **5473.8** |
| halfcheetah-medium-replay | 12135.0 | 2298.2 | 4828.4 | 4492.1 | 4035.7 (365.4) | 4812.0 (148.7) | 3854.8 (966.3) | **6700.6** | 5640.6 |
| halfcheetah-medium-expert | 12135.0 | 8054.4 | 12765.4 | 4169.4 | 633.2 (2152.9) | **10535.6 (3334.9)** | 9470.3 (4178.9) | 7184.7 | 7750.8 |

## 6.3 Ablation Study

The ablation study aims to illustrate how hyperparameters can be determined according to the property of the offline datasets without online evaluation. In particular, COIL has two critical hyperparameters: $\alpha$ and $N$. Since $\alpha$ determines the rate of filtering out the bad trajectories as the agent imitates a better policy where small $\alpha$ corresponds to a high filtering rate, it is highly influenced by the changes in returns of the trajectories in the dataset. Taking `hopper-medium-replay` and `hopper-medium` for examples, the former consists of online training experience, and the return of trajectories changes rapidly; on the contrary, most of the trajectories in the latter are at the same return level. Therefore, intuitively, a small value of $\alpha$ should be assigned to `hopper-medium-replay` while a large value to `hopper-medium`, as confirmed by the experimental results shown in Fig. 6. Ablation experiments on the other hyperparameter $N$ are

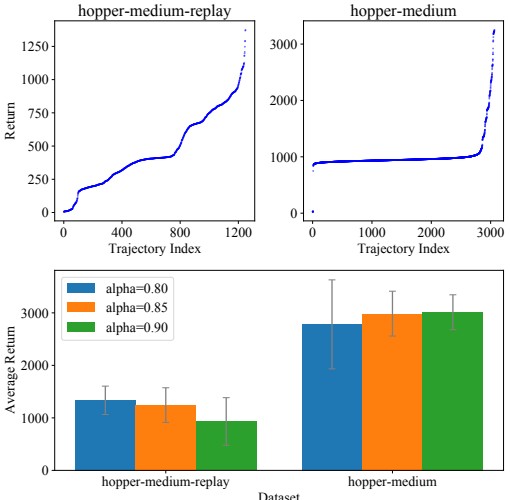

Figure 6: Returns of trajectories in `hopper-medium-replay` and `hopper-medium`, and final performances of COIL with different $\alpha$.

illustrated in Appendix E, saying that it can also be tuned according to the characterization of the dataset.

## 6.4 Compared with Naive Curriculum Strategies

In this section, we propose comparing the curriculum strategy of COIL with naive curriculum strategies to determine whether COIL is a better choice. The first strategy is called *return-ordered BC*, which conducts curriculum imitation learning based only on returns of the trajectories. Similar to COIL, it picks $N_{RBC}$ trajectories with the lowest returns for each curriculum to perform behavioral cloning, and then removes them from the dataset. When the dataset is empty, the algorithm stops. The other strategy is called *buffer-shrinking BC*. As its name suggests, it conducts curriculum imitation learning by shrinking the dataset ordered by the return after training at each curriculum. In detail, *buffer-shrinking BC* begin its training with the entire dataset in the buffer; after a fixed number of

gradient steps, it shrinks the buffer by discarding $p\%$ of trajectories with the lowest returns; then the training is continued on the remaining trajectories. In our experiment, we choose $p = 20$ so that the algorithm will stop after 5 times of shrinkage.

The learning results on *final* datasets are shown in Fig. 7. As expected, *return-ordered BC* leads to mediocre behaviors since similar returns do not always account for similar policies due to the exploration noise. In addition, *buffer-shrinking BC* is not usually stable to achieve the optimal behavior. Since the shrink strategy is totally hand-crafted, the *quantity-quality dilemma* is not eliminated. The mediocre trajectories in the last curriculum will lead to the failure. On the contrary, COIL succeeds in the optimal behavior policy with the highest training efficiency (the least gradient steps), indicating the advantage of the policy-distance-based curriculum.

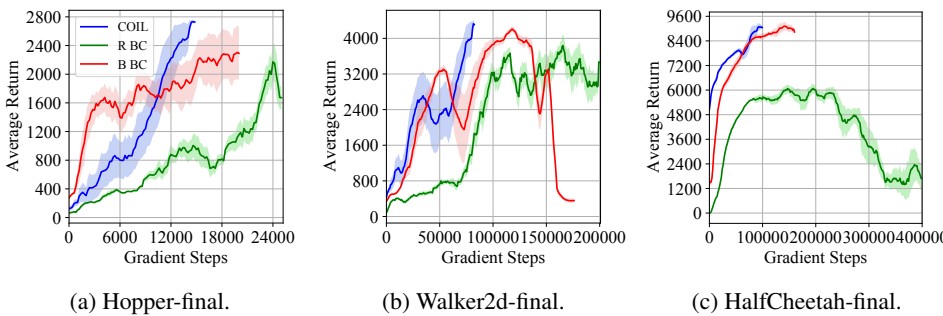

| (a) Hopper-final. | (b) Walker2d-final. | (c) HalfCheetah-final. |

Figure 7: Comparison of training curves between COIL and Return-ordered BC (*R BC*) and Buffer-shrinking BC (*B BC*) on *final* datasets with the same batch size. Different strategies terminate with different gradient steps.

# 7   Conclusion

In this paper, we analyze the quantity-quality dilemma of behavior cloning (BC) from both an experimental and a theoretical point of view, which motivates us to propose the curriculum offline imitation learning (COIL). COIL takes advantage of imitation learning by improving the current policy with adaptive neighboring policies. Experiments show good properties of COIL with competitive evaluation results against state-of-the-art offline RL algorithms. COIL may provide a practical way for bringing offline RL into practice due to its simplicity and effectiveness, but it also limits into the performance of the dataset.

# Acknowledgements

The corresponding author Weinan Zhang is supported by "New Generation of AI 2030" Major Project (2018AAA0100900), Shanghai Municipal Science and Technology Major Project (2021SHZDZX0102) and National Natural Science Foundation of China (62076161, 61632017). The author Minghuan Liu is supported by Wu Wen Jun Honorary Doctoral Scholarship, AI Institute, Shanghai Jiao Tong University. We sincerely thank the reviewers for helpful feedback.

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
