# Appendices

## A   Algorithm

---

**Algorithm 1** Curriculum Offline Imitation Learning (COIL)

---

**Require:** Offline dataset $\mathcal{D}$, number of trajectories picked at each curriculum $N$, moving window of the return filter $\alpha$, number of training iteration $L$, batch size $B$, number of pre-train times $T$, and the learning rate $\eta$.
  Initialize policy $\pi$ with random parameter $\theta$.
  Initialize the return filter $V = 0$.
  **if** $\mathcal{D}$ is collected by a single policy **then**
    Do pre-training for $T$ times using BC.
  **end if**
  **while** $\mathcal{D} \neq \emptyset$ **do**
    **for all** $\tau_i \in \mathcal{D}$ **do**
      Calculate $\tau_i(\pi) = \{\pi(a_0^i|s_0^i), \pi(a_1^i|s_1^i), \cdots, \pi(a_h^i|s_h^i)\}$.
      Sort $\tau_i(\pi)$ into $\{\pi(\bar{a}_0^i|\bar{s}_0^i), \pi(\bar{a}_1^i|\bar{s}_1^i), \cdots, \pi(\bar{a}_h^i|\bar{s}_h^i)\}$ in an ascending order, such that $\pi(\bar{a}_j^i|\bar{s}_j^i) \leq \pi(\bar{a}_{j+1}^i|\bar{s}_{j+1}^i), \quad j \in [0, h-1]$
      Choose $s(\tau_i) = \pi(\bar{a}_{\lfloor \beta h \rfloor}^i|\bar{s}_{\lfloor \beta h \rfloor}^i)$ as the criterion of $\tau_i$.
    **end for**
    Select $N = \min\{N, |\mathcal{D}|\}$ trajectories $\{\bar{\tau}\}_1^N$ with the highest $s(\tau)$ as a new curriculum.
    Initialize a new replay buffer $\mathcal{B}$ with $\{\bar{\tau}\}_1^N$.
    $\mathcal{D} = \mathcal{D} \backslash \{\bar{\tau}\}_1^N$.
    **for** $n = 1 \rightarrow L \times N$ **do**
      Draw a random batch $\{(s, a)\}_1^B$ from $\mathcal{B}$.
      Update $\pi_\theta$ using behavior cloning

$$\theta \leftarrow \theta - \eta \nabla_\theta \sum_{j=1}^{B} \left[ -\log \pi_\theta(a_j|s_j) \right]$$

    **end for**
    Update the return filter $V \leftarrow (1-\alpha)V + \alpha \cdot \min\{R(\bar{\tau})\}_1^N$.
    Filter $\mathcal{D}$ by $\mathcal{D} = \{\tau \in \mathcal{D} \mid R(\tau) \geq V\}$.
  **end while**

---

## B   Proofs

### B.1   Proof for Theorem 1

We introduce useful lemmas before providing our proof.

**Lemma 1** (Total variation distance of joint distributions). *Given two joint distributions $\rho_1(x, y) = \rho_1(x|y)\rho_1(y)$ and $\rho_2(x, y) = \rho_2(x|y)\rho_2(y)$, then their total variation distance can be bounded by*

$$D_{\mathrm{TV}}(\rho_1(x,y)\|\rho_2(x,y)) \leq D_{\mathrm{TV}}(\rho_1(y)\|\rho_2(y)) + \mathbb{E}_{y \sim \rho_2}\left[D_{\mathrm{TV}}(\rho_1(x|y)\|\rho_2(x|y))\right]$$

*Proof.* Directly extend the left hand side, we have

$$D_{\text{TV}}(\rho_1(x,y)\|\rho_2(x,y)) = \frac{1}{2}\sum_{x,y}|\rho_1(x,y) - \rho_2(x,y)|$$

$$= \frac{1}{2}\sum_{x,y}|\rho_1(x|y)\rho_1(y) - \rho_2(x|y)\rho_2(y)|$$

$$= \frac{1}{2}\sum_{x,y}|\rho_1(x|y)(\rho_1(y) - \rho_2(y)) + (\rho_1(x|y) - \rho_2(x|y))\rho_2(y)|$$

$$\leq \frac{1}{2}\sum_{x,y}\rho_1(x|y)|(\rho_1(y) - \rho_2(y))| + \frac{1}{2}\sum_{x,y}|(\rho_1(x|y) - \rho_2(x|y))|\rho_2(y)$$

$$= \frac{1}{2}\sum_{y}|(\rho_1(y) - \rho_2(y))| + \frac{1}{2}\sum_{x,y}|(\rho_1(x|y) - \rho_2(x|y))|\rho_2(y)$$

$$= D_{\text{TV}}(\rho_1(y)\|\rho_2(y)) + \mathbb{E}_{y\sim\rho_2(y)}[D_{\text{TV}}(\rho_1(x|y)\|\rho_2(x|y))]$$

$$\square$$

Therefore, we have the following proposition.

**Proposition 1.**

$$D_{\text{TV}}(\rho_\pi(s,a)\|\rho_{\pi_b}(s,a)) \leq D_{\text{TV}}(\rho_\pi(s)\|\rho_{\pi_b}(s)) + \mathbb{E}_{s\sim\rho_{\pi_b}(s)}[D_{\text{TV}}(\pi(a|s)\|\pi_b(a|s))] \quad (10)$$

*Proof.* Applying Lemma 1 to $\rho_\pi(s,a) = \rho_\pi(s)\pi(a|s)$ and $\rho_{\pi_b}(s,a) = \rho_{\pi_b}(s)\pi_b(a|s)$ and completes the proof. $\square$

**Lemma 2** (Lemma 5.1 of Xu et. al [24])**.** *Let $\Pi$ be the set of all deterministic policy and $|\Pi| = |\mathcal{A}|^{|\mathcal{S}|}$. Assume that there does not exist a policy $\pi \in \Pi$ such that $\pi(s^i) = a^i, \forall i \in \{1, \cdots, |\mathcal{D}|\}$. Then, for any $\delta \in (0,1)$, with probability at least $1 - \delta$, the following inequality holds:*

$$\mathbb{E}_{s\sim\rho_{\pi_b}(s)}[D_{\text{TV}}(\pi(a|s)\|\pi_b(a|s))] \leq \frac{1}{|\mathcal{D}|}\sum_{i=1}^{|\mathcal{D}|}\mathbb{I}\left[\pi(s^i) \neq a^i\right] + \left[\frac{\log|\Pi| + \log(2/\delta)}{2|\mathcal{D}|}\right]^{\frac{1}{2}} \quad (11)$$

We are now ready to give the proof for Theorem 1.

**Theorem 1** (Performance bound of BC)**.** *Let $\Pi$ be the set of all deterministic policy and $|\Pi| = |A|^{|S|}$. Assume that there does not exist a policy $\pi \in \Pi$ such that $\pi(s^i) = a^i, \forall i \in \{1, \cdots, |\mathcal{D}|\}$. Let $\hat{\pi}_b$ be the empirical behavior policy and the corresponding state marginal occupancy is $\rho_{\hat{\pi}_b}$. Suppose BC begins from initial policy $\pi_0$, and define $\rho_{\pi_0}$ similarly. Then, for any $\delta > 0$, with probability at least $1 - \delta$, the following inequality holds:*

$$D_{\text{TV}}(\rho_\pi(s,a)\|\rho_{\pi_b}(s,a)) \leq \frac{1}{2}\sum_{s\notin\mathcal{D}}\rho_{\pi_b}(s) + \frac{1}{2}\sum_{s\notin\mathcal{D}}|\rho_\pi(s) - \rho_{\pi_0}(s)| + \frac{1}{2}\sum_{s\notin\mathcal{D}}|\rho_{\pi_0}(s) - \rho_{\pi_b}(s)|$$

$$+ \frac{1}{2}\sum_{s\in\mathcal{D}}|\rho_\pi(s) - \rho_{\hat{\pi}_b}(s)| + \frac{1}{|\mathcal{D}|}\sum_{i=1}^{|\mathcal{D}|}\mathbb{I}\left[\pi(s^i) \neq a^i\right] + \left[\frac{\log|\mathcal{S}| + \log(2/\delta)}{2|\mathcal{D}|}\right]^{\frac{1}{2}} + \left[\frac{\log|\Pi| + \log(2/\delta)}{2|\mathcal{D}|}\right]^{\frac{1}{2}}$$

$$(12)$$

*Proof.* By Lemma 2, for any $\delta \in (0,1)$, with probability at least $1 - \delta$, the second term in Eq. (10) is bounded by:

$$\mathbb{E}_{s\sim\rho_{\pi_b}(s)}[D_{\text{TV}}(\pi(a|s)\|\pi_b(a|s))] \leq \frac{1}{|\mathcal{D}|}\sum_{i=1}^{|\mathcal{D}|}\mathbb{I}\left[\pi(s^i) \neq a^i\right] + \left[\frac{\log|\Pi| + \log(2/\delta)}{2|\mathcal{D}|}\right]^{\frac{1}{2}} \quad (13)$$

Then we focus on the first term in Eq. (10). Introducing a new distribution $\rho_{\hat{\pi}_b}(s)$, the triangle inequality goes that:

$$D_{\mathrm{TV}}(\rho_\pi(s)\|\rho_{\pi_b}(s)) \leq D_{\mathrm{TV}}(\rho_\pi(s)\|\rho_{\hat{\pi}_b}(s)) + D_{\mathrm{TV}}(\rho_{\hat{\pi}_b}(s)\|\rho_{\pi_b}(s))$$

$$= \frac{1}{2}\sum_{s\in\mathcal{S}} |\rho_\pi(s) - \rho_{\hat{\pi}_b}(s)| + D_{\mathrm{TV}}(\rho_{\hat{\pi}_b}(s)\|\rho_{\pi_b}(s))$$

$$= \frac{1}{2}\sum_{s\notin\mathcal{D}} \rho_\pi(s) + \frac{1}{2}\sum_{s\in\mathcal{D}} |\rho_\pi(s) - \rho_{\hat{\pi}_b}(s)| + D_{\mathrm{TV}}(\rho_{\hat{\pi}_b}(s)\|\rho_{\pi_b}(s))$$

$$\leq \frac{1}{2}\sum_{s\notin\mathcal{D}} \rho_{\pi_b}(s) + \frac{1}{2}\sum_{s\notin\mathcal{D}} |\rho_\pi(s) - \rho_{\pi_b}(s)| + \frac{1}{2}\sum_{s\in\mathcal{D}} |\rho_\pi(s) - \rho_{\hat{\pi}_b}(s)| \quad (14)$$

$$+ D_{\mathrm{TV}}(\rho_{\hat{\pi}_b}(s)\|\rho_{\pi_b}(s))$$

$$\leq \frac{1}{2}\sum_{s\notin\mathcal{D}} \rho_{\pi_b}(s) + \frac{1}{2}\sum_{s\notin\mathcal{D}} |\rho_\pi(s) - \rho_{\pi_0}(s)| + \frac{1}{2}\sum_{s\notin\mathcal{D}} |\rho_{\pi_0}(s) - \rho_{\pi_b}(s)|$$

$$+ \frac{1}{2}\sum_{s\in\mathcal{D}} |\rho_\pi(s) - \rho_{\hat{\pi}_b}(s)| + D_{\mathrm{TV}}(\rho_{\hat{\pi}_b}(s)\|\rho_{\pi_b}(s))$$

Denote $\mathcal{S}_\mathcal{D} = \{s \mid s \in \mathcal{D}\}$. Noticing that $\mathbb{E}_{\mathcal{S}_\mathcal{D}\sim\rho_{\pi_b}(s)}[\hat{\rho}_{\hat{\pi}_b}(s)] = \rho_{\pi_b}(s)$, by union bound and Hoeffding's inequality, the following inequality holds:

$$P[D_{\mathrm{TV}}(\rho_{\hat{\pi}_b}(s)\|\rho_{\pi_b}(s)) > \epsilon] = P[\exists s \in \mathcal{S}, |\rho_{\hat{\pi}_b}(s) - \rho_{\pi_b}(s)| > \epsilon]$$

$$\leq \sum_{s\in\mathcal{S}} P[|\rho_{\hat{\pi}_b}(s) - \rho_{\pi_b}(s)| > \epsilon] \quad (15)$$

$$\leq 2|\mathcal{S}|e^{-2|\mathcal{D}|\epsilon^2}$$

Let $\delta$ be the right hand side, we obtain that with probability at least $1 - \delta$, $D_{\mathrm{TV}}(\rho_{\hat{\pi}_b}(s)\|\rho_{\pi_b}(s))$ is bounded by:

$$D_{\mathrm{TV}}(\rho_{\hat{\pi}_b}(s)\|\rho_{\pi_b}(s)) \leq \left[\frac{\log|\mathcal{S}| + \log(2/\delta)}{2|\mathcal{D}|}\right]^{\frac{1}{2}} \quad (16)$$

Combining Proposition 1, Ineq. (13), Ineq. (14) and Ineq. (16) completes the proof. $\square$

## B.2 Proof for Observation 1

**Observation 1.** *Under the assumption that each trajectory $\tau_{\tilde{\pi}}$ in the dataset $\mathcal{D}$ is collected by an unknown deterministic behavior policy $\tilde{\pi}$ with an exploration ratio $\beta$. The requirement of the KL divergence constraint $\mathbb{E}_{\tilde{\pi}}\left[D_{KL}(\tilde{\pi}(\cdot|s)\|\pi(\cdot|s))\right] \leq \epsilon$ suffices to finding a trajectory that at least $1 - \beta$ state-action pairs are sampled by the current policy $\pi$ with a probability of more than $\epsilon_c$ such that $\epsilon_c \geq 1/\exp\epsilon$, i.e.:*

$$\mathbb{E}_{(s,a)\in\tau_{\tilde{\pi}}}[\mathbb{I}(\pi(a|s) \geq \epsilon_c)] \geq 1 - \beta , \quad (17)$$

*Proof.* We begin with the KL divergence constraint:

$$\mathbb{E}_{\tilde{\pi}}\left[D_{\mathrm{KL}}(\tilde{\pi}(\cdot|s)\|\pi(\cdot|s))\right] \leq \epsilon$$

$$\Rightarrow \qquad \mathbb{E}_{\tilde{\pi}}\left[\log\frac{\tilde{\pi}(a|s)}{\pi(a|s)}\right] \leq \epsilon$$

$$\Rightarrow \qquad \mathbb{E}_{(s,a)\in\tau_{\tilde{\pi}}}\left[\log\pi(a|s)\right] \geq \log(1-\beta) - \epsilon \quad (18)$$

$$\Rightarrow \qquad \log\mathbb{E}_{(s,a)\in\tau_{\tilde{\pi}}}\left[\pi(a|s)\right] \geq \log(1-\beta) - \epsilon$$

$$\Rightarrow \qquad \mathbb{E}_{(s,a)\in\tau_{\tilde{\pi}}}\left[\pi(a|s)\right] \geq \frac{(1-\beta)}{\exp\epsilon}$$

Besides, to achieve $\mathbb{E}_{(s,a)\in\tau_{\tilde{\pi}}}[\mathbb{I}(\pi(a|s) \geq \epsilon_c)] \geq 1 - \beta$, for any state-action pair in $\tau_{\tilde{\pi}}$, there are at least $1 - \beta$ of them can be sampled by $\pi$ with at least the probability of $\epsilon_c$. Therefore, we have the following lower bound:

$$\mathbb{E}_{(s,a)\in\tau_{\tilde{\pi}}}[\pi(a|s)] \geq (1-\beta)\cdot\epsilon_c + 0\cdot\beta = (1-\beta)\cdot\epsilon_c \quad (19)$$

Combing Ineq. (18) and Ineq. (19) completes the proof.

$\square$

# C Implementation Details

## C.1 Implementation for Estimating the Empirical Discrepancy

In this part we explain how we estimate the empirical discrepancy outside the dataset, *i.e.*, the term $\frac{1}{2}\sum_{s\notin\mathcal{D}}|\hat{\rho}_{\pi}(s) - \hat{\rho}_{\pi_0}(s)|$ in Theorem 1.

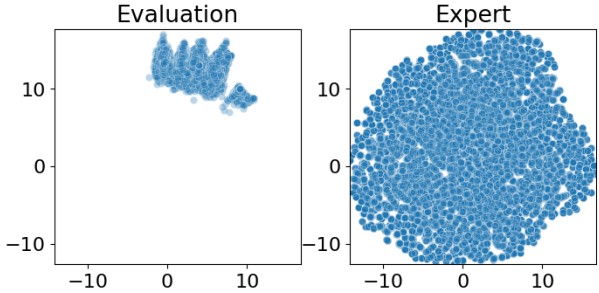

Figure 8: An illustration to the UMAP transformation. Left: states sampled by a random policy in 2-dimensional space transformed by UMAP. Right: expert data to train UMAP in 2-dimensional space.

Due to the difficulty of estimating the empirical state marginal occupancy measure $\hat{\rho}_{\pi}(s)$ for a policy $\pi$, especially in continuous or high-dimensional tasks, we estimation its value in 2-dimensional space by projection.

In detail, we choose UMAP [15] as the projection algorithm and train the projecting function in Hopper using $64\,000$ transitions sampled by the expert agent. To evaluate a policy, we sample the same number of transitions, and then project them onto a 2-dimensional space by the trained projecting function. Fig. 8 illustrates the projected state distribution, where the left denote the state sampled by a random policy, and the right is the expert data.

For empirical estimation, we subsequently discretize the projected 2-dimensional state space into small grid regions, and estimated the distribution via Kernel Density Estimation (KDE) [19] with Gaussian kernel. Suppose $g_{\pi}(\Delta)$ is the Gaussian kernel density function of the transformed data sampled by policy $\pi$, and the dataset is $\mathcal{D}$. Denote the region that contains state $s$ as $\Delta(s)$. Then the empirical discrepancy can be approximated by

$$\frac{1}{2}\sum_{s\neq D}|\hat{\rho}_{\pi}(s) - \hat{\rho}_{\pi_0}(s)| \approx \frac{1}{2}\sum_{\text{all }\Delta}|g_{\pi}(\Delta) - g_{\pi_0}(\Delta))| - \frac{1}{2}\sum_{s\in\mathcal{D}}|g_{\pi}(\Delta(s)) - g_{\pi_0}(\Delta(s))| \quad (20)$$

## C.2 Implementation for the Main Experiment

The implementation of COIL and BC are based on a Pytorch code framework[4]. As for compared baselines, we take their official implementation:

- CQL [13]: `https://github.com/aviralkumar2907/CQL`

- BAIL [2]: `https://github.com/lanyavik/BAIL`

- AWR [17]: `https://github.com/xbpeng/awr`

It should be noted that the public code of AWR is for online RL tasks and we have to modify the code to obtain an offline version, following the instructions in their paper [17].

Table 3: Important Hyperparameters.

rd: random   mr: medium-replay   md: medium   me: medium-expert

| Environments | Hopper | | | | | Walker2d | | | | | HalfCheetah | | | | |
|---|---|---|---|---|---|---|---|---|---|---|---|---|---|---|---|
| | final | rd | mr | md | me | final | rd | mr | md | me | final | rd | mr | md | me |
| Optimizer | AdamOptimizer | | | | | | | | | | | | | | |
| Discount factor $\gamma$ | 0.99 | | | | | | | | | | | | | | |
| Batch size | 256 | | | | | | | | | | | | | | |
| Tuning range of the filter window size $\alpha$ | [0.8, 0.85, 0.9] | | | | | | | | | | | | | | |
| Tuning range of the number of selected trajectories $N$ | [1,2] | | | | | | | | | [5,10] | [1,2,5] | | | | [5,10] |
| Tuning range of $L$ | [50, 100, 200] | | | | | | | | | | [100, 200, 400] | | | | |
| Tuning range of $\eta_\pi$ | [3e-5, 1e-4] | | | | | | | | | | | | | | |

# D   Important Hyperparameters

The main hyperparameters used in our experiments are shown in Tab. 3. Based on the evaluation results of the terminating policy of COIL, might be helpful guidelines for utilizing COIL with different tuning choices are as follows:

**Policy learning rate $\eta_\pi$.** Similar to CQL [13], we evaluated COIL with a policy learning rate in the range of $[3e-5, 1e-4]$. We find that $1e-4$ almost uniformly attain good performance and we chose $1e-4$ as the default across all datasets. Besides, we recommend increasing the number of gradient steps $L$ to be compatible with a low learning rate on the same task.

**Gradient steps for each curriculum $L$.** The gradient steps correspond to how long that BC should be utilized for imitating the target policy at the current level. For easier tasks as Hopper and Walker, we evaluate COIL in the range of $[50, 100, 200]$ and for harder tasks like Halfcheetah, we tune in the range of $[100, 200, 400]$. The default choice is $100$.

**Number of selected trajectories $N$ and Filter window size $\alpha$.** These two hyperparameters affect the experimental results more significantly. Moreover, as mentioned in Section 6.3, they can be tuned based on the distribution of the dataset. Detailed guidelines can be found in Section 6.3 and Appendix E.2.

# E   Additional Results

## E.1   Results of the Motivating Example

Here we provide additional results for the motivated example experiments performed on the Hopper environment.

Table 4: Numerical values of results presented in Fig. 2b, the means and standard deviations are calculated over 3 random seeds.

| Agent | Initial | 1 Trajectory | 4 Trajectories | 256 Trajectories | 1024 Trajectories |
|---|---|---|---|---|---|
| Random | 49.9 (3.0) | 255.0 (355.1) | 374.1 (392.8) | 2802.9 (846.4) | 3004.7 (656.4) |
| 1/3 Return | 1362.6 (853.7) | 2930.3 (886.9) | 2709.5 (974.2) | 3096.4 (630.9) | 3310.2 (442.0) |
| 1/3 Trained | 3289.1 (3.43) | 3014.4 (926.0) | 3428.4 (392.9) | 3582.2 (115.9) | 3588.3 (163.7) |
| 2/3 Trained | 3558.0 (6.0) | 3609.8 (14.2) | 3602.6 (80.3) | 3612.4 (78.8) | 3610.6 (13.9) |
| Demonstration | 3622.3 (22.0) | - | - | - | - |

Table 5: Numerical values of results presented in Fig. 2c, the means and standard deviations are calculated over 3 random seeds.

| Agent | 1 Trajectory | 4 Trajectories | 256 Trajectories |
|---|---|---|---|
| Random | 0.706 (0.111) | 0.737 (0.124) | 0.518 (0.077) |
| 1/3 Trained | 0.528 (0.006) | 0.549 (0.007) | 0.340 (0.004) |
| 2/3 Trained | 0.541 (0.001) | 0.551 (0.006) | 0.313 (0.005) |

We do not evaluate the empirical discrepancy in the case of 1024 trajectories because the projected points are so dense that the grid size is required to be rather small, which leads to an unacceptable computing time.

---

[4]`https://github.com/Ericonaldo/ILSwiss`

## E.2 Additional Ablation Studies

### E.2.1 Ablation on Trajectory Number $N$

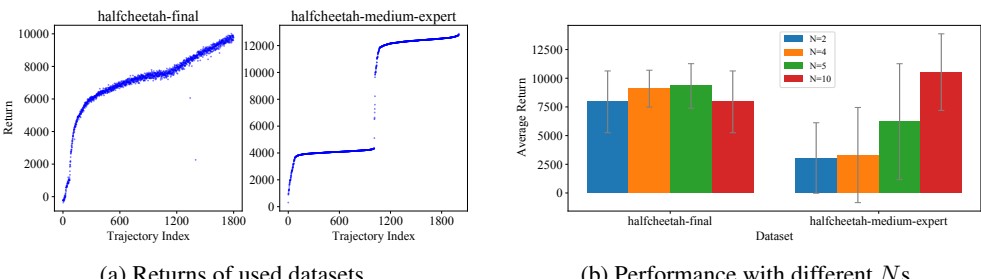

(a) Returns of used datasets.

(b) Performance with different $N$s.

Figure 9: Returns of trajectories in `halfcheetah-final` and `halfcheetah-medium-expert`, and final performances of COIL on them with different $N$.

We further show how the number of chosen trajectories $N$ can be determined by the dataset. Fig. 9b shows the results on `halfcheetah-final` and `halfcheetah-medium-expert` for examples. Obviously, trajectories in `halfcheetah-final` are densely and smoothly arranged than those in `halfcheetah-medium-expert`, indicating that the discrepancy between the behavior policies contained in the final dataset may be smaller. As revealed in Theorem 1, as the distance between the behavior policies becomes farther, more training samples are required for a good imitation. Therefore, a larger $N$ should be chosen for `halfcheetah-medium-expert` than the other one. Fig. 9b shows consistent results to our expectation, where a large value of $N = 10$ puts the best performance on `halfcheetah-medium-expert`, and medium values of $N$ (4 or 5) provide better behaviors on `halfcheetah-final`.

### E.2.2 Further Ablation on Return Filter

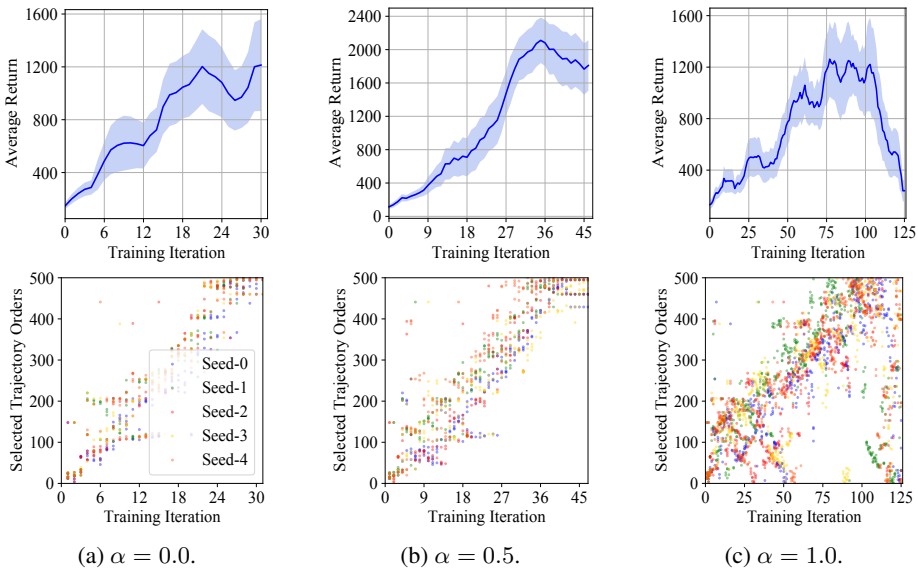

(a) $\alpha = 0.0$.

(b) $\alpha = 0.5$.

(c) $\alpha = 1.0$.

Figure 10: Training curves and orders of selected trajectories with different $\alpha$ on `hopper-final`.

To further illustrate the functionality of the return filter we conduct more ablation experiments on the hyperparameter $\alpha$, where we set $\alpha$ as 1.0 (no return filter), 0.0 (no moving average) and 0.5 (rapid moving average). Obviously, without a return filter ($\alpha$=1.0), the agent imitates earlier trajectories in the final which deteriorates the final performance; without a moving average ($\alpha$=1.0), the agent quickly drops the candidate trajectories which leads it to learn nothing; the results with a rapid moving average ($\alpha$=0.5), are better and more stable but it still fails to imitate the best behavior data.

Therefore, the return filter is a key ingredient for COIL that should be designed carefully when applied on different datasets.

## E.3 Complete Training Curves on Final Datasets

We show the complete training curves of COIL, CQL, AWR and BAIL on *final* datasets. We do not cover D4RL benchmarks since those numerical results of baselines are directly borrowed from Fu et al. [5]. As is observed in Fig. 11, CQL works well on Halfcheetah; but on Hopper and Walker, the other imitation-based methods are more effective to reach a better performance. Saliently, COIL only needs fewer gradient steps to terminate with an excellent policy, such that we have to use a different scale of axis (the top axis) to illustrate COIL clearly.

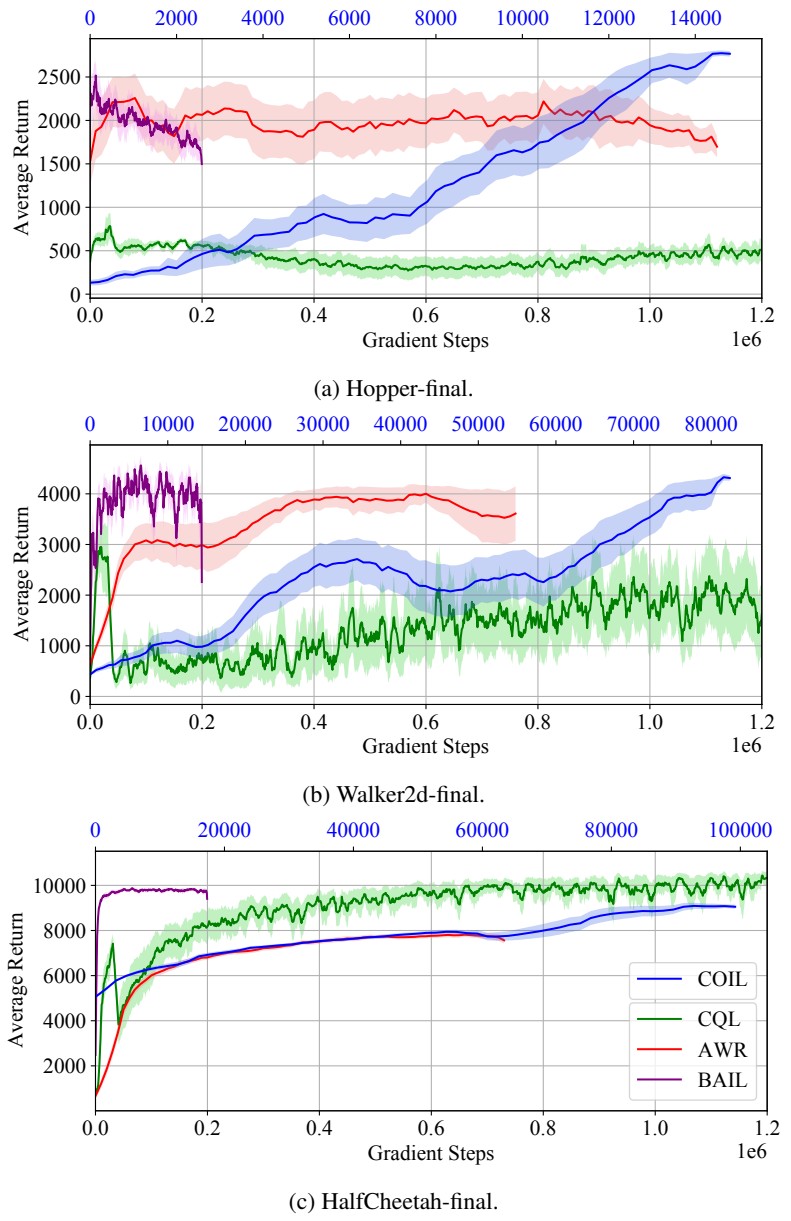

(a) Hopper-final.

(b) Walker2d-final.

(c) HalfCheetah-final.

Figure 11: Comparison of training curves between COIL, CQL, AWR, and BAIL on *final* datasets. Except BAIL has a large batch size (1000), the other methods keep the same batch size (256). Different methods terminate with different gradient steps. The top axis (blue) of each figure illustrates the gradient step of COIL in a small magnitude, showing the highest efficiency of our method. And the bottom axis (black) denotes the gradient step for the other baselines.