# OpenReview forum: "Curriculum Offline Imitating Learning"
_NeurIPS.cc/2021/Conference — NeurIPS 2021 Poster_

### Official Review · Reviewer_FWCF · 2021-07-01

**Rating:** 7
**Confidence:** 3

**Summary:**

The method proposes an algorithm for Offline Reinforcement Learning. The work starts from the idea that behavioral cloning struggles at learning from data coming from several policies but, on the contrary, easily learns a behavior policy if initialized with a “close” policy.
A curriculum is thus built for behavioral cloning to efficiently learn the “best policy shown in the dataset”. For this, a behavioral cloning is trained on a sequence of trajectories defined the following way: the next trajectory selected for training is the most probable according to current policy. The trajectories-dataset is then filtered by saying that the return should be above a threshold given by a running average of the trajectories seen so far.


**Limitations And Societal Impact:**

No direct societal impact.

**Main Review:**


I appreciate the simplicity of the method, although some explanations could, in my opinion, be simplified. I have questions regarding the experiments although the results are quite convincing.

Remarks:
- 3.1: It would have been more informative to take the ⅓ (resp. ⅔ ) policy at ⅓ of the final performance rather than ⅓ of the training steps. The initializations are quite close in terms of return so I find it quite normal that such a good inits lead to great final performance.

Derivation of the algorithm:

- I find the derivation of the algorithm a bit confusing. First, there are some mistakes in the mathematical notations that make reading harder (e.g. l.213, no consistency between $(s_t, a_t)$ and $(s, a)$) or in the Eq.7 where there are confusion between $\pi_i$ and $\pi$, as well as between $\hat{\pi}_i$ and  $\hat{\pi}$.

- Proposition 1) is not rigorous. “Corresponds to” does not mean anything mathematically. The proof in B.2 uses implications rather than equivalences. So you proved that a trajectory satisfying the "KL condition" also satisfies the “probability condition”. However you are using the converse property.

- However, the final way of defining a “close policy” is simple and makes sense. I would advise either cutting off the KL based argumentation and go directly with the heuristic or refine the mathematical derivations.

- Even though the return filter is a very simple idea, I very much appreciate the fact that it brings a natural termination condition to the algorithm. I would like to see an ablation without the return filter at least in the appendix.

Experiments
- I argue that results of Table 1 should not appear as is. Indeed, when introducing a new dataset, it is unfair for baselines to use “default hyperparameters”. If you introduce a new dataset, you need to sweep the hyperparameters of the baselines at least as much as the ones of your own algorithm (or at the very least the learning rate...). You also need to specify how hyperparameter selection was performed. For your information, [1] discussed how to select HPs in offline RL and [2] discussed other ways to perform HP selection for imitation.

- Otherwise, the results are quite well presented and convincing.
Baselines in appendix E.2 are very natural and I would recommend adding it to the core of the paper if you have the space to do so. Can you however explain the weird x-axis with algorithms not all starting at step 0?

-Ablation E.3 is also very insightful. It is very helpful to assist potential users in finding good hyperparameters.

It is overall a good paper. I would be ready to increase my score if authors clarify the algorithm mathematical derivation and either solve the experimental issues I mentioned or debunk my arguments.

[1] Paine, Tom Le, et al. "Hyperparameter selection for offline reinforcement learning." arXiv preprint arXiv:2007.09055 (2020).
[2] Hussenot, Leonard, et al. "Hyperparameter Selection for Imitation Learning." arXiv preprint arXiv:2105.12034 (2021).




-----------    UPDATE AFTER AUTHOR RESPONSE  -------------

Thanks for the very detailed answers!

Q1: Thanks for adding this additional datapoint. I am glad to see that this 1/3 performance point also confirms your hypothesis.

Q3: It is a shame we cannot update the pdf, but I am happy the authors will now present the KL derivations as a relaxation rather than an equivalence. This will make the paper more rigorous and will not diminish the impact of its results.

Q4: Thank you for adding this ablation. It looks confusing in the authors response (typo in the reported numbers?) but the figure that was shared is clear and supporting the importance of the return filter.

Q5: Thanks for the additional efforts tuning the baselines. Please include these new results in the main paper.

Q7: I would recommend aligning the beginning of the training rather than the end... otherwise the x-axis is just.... wrong.

Given the additional work, including results and clarifications, provided by the authors, I increased my score to 7.


**Time Spent Reviewing:**

3

---

> ### Author Response · Authors · 2021-08-08
> **Response to Reviewer FWCF**
>
> We appreciate your constructive and helpful comments and we would like to let you know that it is very useful for us to polish the paper.
>
> **Q1: "It would have been more informative to take the ⅓ (resp. ⅔ ) policy at ⅓ of the final performance rather than ⅓ of the training steps"**
>
> **A1:** We agree with your constructive suggestion, and we in the revision have supplemented the experiment in this section. We save a checkpoint at the early training stage where the performance of the SAC agent is around 1200 (1/3 of the final performance). We then train the policy from the checkpoint using BC as in Section 3.1, and list the numerical results below. The updated version of Figure 2b in our paper can be seen in the external link (https://anonymous.4open.science/r/coil-rebuttal-5334/COIL_rebuttal.pdf).
>
> | number of demo trajs | initial | 1 | 4 | 256 | 1024 |
> |:----------:|:----------:|:----------:|:----------:|:----------:|:----------:|
> | average return | 1362.6±853.7 | 2930.3±886.9 | 2709.5±974.2 | 3096.4±630.9 | 3310.2±442.0 |
>
> **Q2: "..., derivation of the algorithm a bit confusing..."**
>
> **A2:** We apologize for the confusing notation/derivation and we are continuing to polish them following your and the other reviewers' suggestions. The Consistency in l.213 and eq(7) has been solved.
>
> **Q3: "Proposition 1) is not rigorous...However you are using the converse property"**
>
> **A3:** We sincerely thank you for your detailed checking and advice about the non-rigorous proposition. We do find it is not appropriate to claim as such, therefore, we decide to revise this proposition as an observation that relaxes the original KL condition (but is not totally heuristic) to guide us to design the simple yet effective algorithm.
>
> **Q4: "I would like to see an ablation without the return filter at least in the appendix."**
>
> **A4:** Thanks for your valuable suggestion! We have now supplement such ablation experiments. Here we set the alpha three extreme values: 1.0 (no return filter), 0.5 (very rapid moving average), and 0.0 (no moving average). The numerical results are listed below.
>
> - Overall, without a return filter ($\alpha$=1.0), the agent imitates earlier trajectories in the final which deteriorates the final performance;
>
> - without a moving average ($\alpha$=1.0), the agent quickly drops the candidate trajectories which leads it to learn nothing;
>
> - the results with a rapid moving average ($\alpha$=0.5) are better and more stable but it still fails to imitate the best behavior data.
>
> Additional figures containing training curves and oracle orders of picked trajectories are in the above link.
>
> | alpha=0.0 | alpha=0.5 | alpha=1.0 |
> |:----------:|:----------:|:----------:|
> | 949.0±569.0 | 79.3±134.8 | 1483.6±741.5 |
>
> **Q5: "I argue that results of Table 1 should not appear as is."**
>
> **A5:** Thanks for your valuable suggestion on the hyperparameter tuning for baselines. We have performed a grid search to find the best hyper-parameter for these baselines and added the experiment results in the revision. Some of the results are shown below.
> - For AWR, since its open-source code specifies different details of hyperparameters for different environments (not for different datasets), we decide not to tune it and keep the results.
> - For BAIL, as they do not give any search range of their hyperparameters in both the paper and the code, we tune the learning rate in {1e-4, 3e-4, 1e-3}. The results are as follows. The mean performance raises in both hopper-final and walker2d-final, but its variance is still large, making BAIL a less stable method than us.
>
> | | hopper-final | walker-final | half-final |
> |:----------:|:----------:|:----------:|:----------:|
> | lr=1e-4 | 1942.5±1007.8 |  4081.8±1469.7 | 9503.3±708.7 |
> | lr=3e-4 | 2296.9±915.9 | 4236.2±1531.1 | 9730.8±675.5 |
> | lr=1e-3 |1517.7±932.2 | 2784.5±2381.3 | 9745.0±880.3 |
>
> - For CQL, we tune it as its paper suggests and show the results below ("plr" is the abbreviation of "policy learning rate"). Specifically, the guideline in their code recommends searching alpha in {5.0, 10.0} and the policy learning rate in {1e-4, 3e-4}. There is no significant difference between the best results and our reported results.
>
> | | hopper-final | walker-final | half-final |
> |:----------:|:----------:|:----------:|:----------:|
> | alpha=5.0, plr=1e-4 |41.4±164.0 |2247.5±1986.8 |10550.4±1373.4 |
> | alpha=5.0, plr=3e-4 |207.7±429.7 |1374.7±1928.1 |10528.8±1474.5 |
> | alpha=10.0, plr=1e-4|501.5±227.5 |2604.3±1937.6 |10882.0±1042.7 |
> | alpha=10.0, plr=3e-4|69.2±182.6 |587.0±1331.3 |10753.4±840.3 |
>
> **Q6: "Baselines in Appendix E.2 are very natural and I would recommend adding it to the core of the paper if you have the space to do so."**
>
> **A6:** We appreciate very much for your detailed review. However, due to the space limit, it seems that we can hardly put it into the main text. But we will revise the statement in section 6 to remind the readers that we have more interesting comparison results against naive curriculum strategies in the appendix.
>
> **Q7: "Can you however explain the weird x-axis with algorithms not all starting at step 0?"**
>
> **A7:** We are sorry for the confusion. In fact, as noted in the figure caption. Since these methods terminate at different steps, we align them by the end of the training in order to highlight our higher training efficiency. Here we refer to the paper of BAIL [1], which adopts the same plotting manner.
>
> **References**:
>
> [1]: Xinyue Chen,  Zijian Zhou,  Zheng Wang,  Che Wang,  Yanqiu Wu,  and Keith Ross.   BAIL: Best-action imitation learning for batch deep reinforcement learning. NeurIPS 2020 arXiv:1910.12179, 2019.

---

> ### Author Response · Authors · 2021-08-12
> **Update response to Reviewer FWCF**
>
> We thank you again for the constructive review. And there's something new you may want to know:
> - It is true and we apologize for making a typo that confuses you in the response: we misplace the results of **alpha=0.5** and **alpha=1.0**, as you might have realized according to the figure.  The correct reported numbers in **A4** are:
> | alpha=0.0 | alpha=0.5 | alpha=1.0 |
> |:----------:|:----------:|:----------:|
> | 949.0±569.0 | 1483.6±741.5 | 79.3±134.8 |
> - New results will no doubt be put into the paper.
> - We will take your advice to modify the weird x-axis by aligning the beginning of the training.

---

### Official Review · Reviewer_8Yop · 2021-07-16

**Rating:** 6
**Confidence:** 4

**Summary:**

This paper proposed and interesting algorithm to do imitation learning with an automatic curriculum learning technique. In particular an automatic filtering of better trajectories under the current policy  and a selection of the best ones in that set.

**Limitations And Societal Impact:**

Its seems not readily applicable, but the authors could have written about the benefits of doing Offline RL in the sense of the impact of training. Or a computational analysis could have interesting, in the sense of cost of training, CO2 impact, cost, etc.

**Main Review:**




Originality:  The idea is original, even though it sounds like BAIL for trajectories. But the math has been done well and it is well supported.

Quality: Claims are supported. I believe some sections / paragraphs would benefit for more clarity. Also it would greatly benefit from more extensive experimentation. Adding more environments.

Clarity: The submission is well organized but incomplete. The notation seems porr and writing need clarity, The figures would benefit from extra information some

Significance: The results are better than SOTA. Event thought in D4RL is not compara with CQL and BAIL, and more environments would benefit the claims.

Notes:
Confused by notation in line 91, unclear how the sum of samples belonged to the BC policy corresponds to a well formed policy.
Line 95, notation is not clear, D was defined as having trajectories but here it is used as having policies.
Line 97, re-used D for indication of tasks.
Line 159, Clarity "the close policy".
Line 181. "kinds of policies" grammar.
Line 195. Poor notation. "The target Dt" .
Figure 5 ( show comparison) needs to improve the graph, the seeds graphs as points are confusing.
Would benefit from adding D4RL don't compare with BAIL and CQL [ or even model based offline rl)


**Time Spent Reviewing:**

3

---

> ### Author Response · Authors · 2021-08-08
> **Response to Reviewer 8Yop**
>
> Thanks for your valuable comments. We commit to improving the clarity of notations/algorithm and details/figures.  We have also added more baselines for better comparison in the revision following your suggestions.
>
> **Q1. Notation errors:**
>
> - **Q1.1: "Confused by notation in line 91, unclear how the sum of samples belonged to the BC policy corresponds to a well formed policy"**
>
> - **A1.1**: We apologize for the confusion. We will add an intuitive explanation in the revision. Intuitively, the assumed behavior policy is a mixed policy that is combined by all the policies collecting such data. And its empirical estimation is the evidence provided by the dataset, which can be seen as constructing a policy from all the data points contained in the dataset.
>
>
> - **Q1.2: "D was defined as having trajectories but here it is used as having policies"**
>
> - **A1.2**: Thanks for your detailed checking. In fact, here we try to characterize the policy from $D$ as the data collection policy. We have revised the statement to ease the confusion as:
>     - ... in a mixed dataset $\mathcal{D}$ that is collected by $K$ different policies $\pi_1, \cdots, \pi_K$, the optimal behavior policy $\pi^*$ can be determined such that for $\forall i\in[1,K],~ \pi_i \preceq \pi^*$.
>
>
> - **Q1.3: "Line 97, re-used D for indication of tasks."**
>
> - **A1.3**: Thanks again for the checking. This is no doubt a typo and, we have fixed the reused notations in our revision.
>
>
> - **Q1.4: "Line 159, Clarity 'the close policy'"**
>
> - **A1.4**: We have changed it to "the neighboring policy".
>
>
> - **Q1.5:  "Line 181. 'kinds of policies' grammar"**
>
> - **A1.5**: We have changed it to "various policies".
>
>
> - **Q1.6: "Line 195. Poor notation. 'The target Dt'".**
>
> - **A1.6**: Thanks again for the checking. The paper actually writes "the target task Dt". And we have fixed the notation errors in our revision.
>
>
>
> **Q2: "Figure 5 (show comparison) needs to improve the graph"**
>
> **A2:** We guess you may misunderstand the meaning of Figure 5, and we will illustrate figure 5 in a better way. Actually, as is said in the paper, Figure 5 does not show any comparison but is trying to analyze what the algorithm does during the training procedure.
>
> - The upper figures show the learning curves of our method on those final buffer datasets.
>
> - The bottom figures show the oracle online trajectory orders (i.e., when does the online SAC agent collect them) of selected trajectories by our method during the training. The points represent the selected trajectories, where the x-axis is the training iteration and the y-axis is the oracle order of the trajectory.
>
> - From our analysis, we find that COIL selects the trajectories following the original training orders, supporting the basic idea of adaptively imitating the neighboring policies, and finally terminating with a near-data-optimal policy.
>
> **Q3: "Would benefit from adding D4RL don't compare with BAIL and CQL (or even model-based offline rl)"**
>
> **A3:** Thanks for your valuable comments. We have run BAIL on D4RL for a better comparison in the revision.
>
> - The additional results of BAIL are listed in the following table. All means and standard deviations are calculated over 5 seeds (or in the external link (https://anonymous.4open.science/r/coil-rebuttal-5334/COIL_rebuttal.pdf).
> - In addition, since D4RL has covered CQL, the SoTA column indeed includes the results of CQL. We will make our statement clear in the revision.
> - As for comparing model-based offline RL methods, we have involved a recent work (MOPO[1]) and the results from their paper in our revision (which can also be seen in the above link). We conclude that model-based methods have the potential to gain better performance than all model-free algorithms on the mixed dataset (e.g., the medium-replay datasets), since there are sufficient data to learn a good environment model; however, on the other dataset, MOPO is not competitive with model-free methods, including ours. Therefore, we think such a comparison may not be the main point of this paper.
>
>
> Results of BAIL on D4RL
>
> | hopper-random | hopper-medium|hopper-medium-replay | hopper-medium-expert |
> |:----------:|:-------------:|:----------:|:-------------:|
> | 318.0±5.1 | 1571.5±900.7 | 808.7±192.5 | 2435.9±1265.2 |
>
> | walker-random | walker-medium| walker-medium-replay | walker-medium-expert |
> |:----------:|:-------------:|:----------:|:-------------:|
> | 130.8±87.2 | 1242.4±1545.7 | 532.9±359.0 | 3633.9±1839.7 |
>
> | half-random | half-medium| half-medium-replay | half-medium-expert |
> |:----------:|:-------------:|:----------:|:-------------:|
> | -96.4±49.7 | 4277.6±564.9 | 3854.8±966.3 | 9470.3±4178.9 |
>
> **Q4: "but the authors could have written about the benefits of doing Offline RL in the sense of the impact of training. Or a computational analysis could have interesting, in the sense of the cost of training, CO2 impact, cost, etc"**
>
> **A4:** Thanks for your valuable suggestion! Since our method is simple yet effective, it would be interesting to highlight its strength in terms of computational cost and CO2 impact.
>
> **Other responses about our contribution and completeness:**
>
> - We would also like to thank you for your positive comments on the originality and mathematical derivation. We are excited about this simple yet effective algorithm. It takes a completely different approach to leverage data generated by less optimal policies, which is to form a sequential curriculum to ease the learning towards the best behavior policy in offline data. Despite being simple, it achieves comparable performance with SOTA algorithms. We hope our algorithm can bring a new perspective to offline RL.
>
> - We find your suggestions on writing and experiments very useful for us to polish our paper. We have clarified the notations/figures/algorithm details and added more baselines in both our response and the updated version (which is unfortunately unobservable to you now) according to your comments. Now we think the paper is highly completed. We hope this can address your concerns on clarity and experiments. We would be happy to make further corrections if necessary and look forward to your feedback. We commit to further improve the clarity of our paper in the revision.
>
> **Reference:**
> [1]: Tianhe Yu, Garrett Thomas, Lantao Yu, Stefano Ermon, James Zou, Sergey Levine, Chelsea Finn, and Tengyu Ma. Mopo: Model-based offline policy optimization. NeurIPS 2020 arXiv:2005.13239, 2020

---

> > ### Comment · Reviewer_8Yop · 2021-08-13
> > **Revision**
> >
> > I have update the score based on your comprehensive response.
> >
> > I see this a as SOTA algorithm for offline RL given that surpass others in certain scenarios.

---

### Official Review · Reviewer_PdUi · 2021-07-16

**Rating:** 7
**Confidence:** 2

**Summary:**

This paper applied Online Imitation Learning to offline RL. The central idea is, at each iteration of training, the proposed approach will sample the data points from the neighboring policies based on the experience picking. The intuition is built based on the observation/analysis of the discrepancy between demo policy and initialized imitating policy.  I like how authors scrutinized their observations with theoretical analysis. The neighboring policy picking targets at solving the "dilemma" of quantity-quality. However, the main technical contribution is not super very in-depth, authors tackle the KL-divergence constraints through the assumption that each trajectory is sampled by a single policy, which I am not sure whether it is very practical as claimed by authors (yield good bound and performance as illustrated in experimental studies).

**Limitations And Societal Impact:**

Yes

**Main Review:**

Strength:
- Paper is very well-written and easy to follow. The authors moved almost all proof to the appendix and made the paper accessible.
- The central ideas are motivated by real-world observations and theoretical analysis

Weakness:
- The experimental evaluation could be done with more complex datasets
- The technical depth: The evaluation of the two policies is based on relaxing the trajectories sampled by a single policy, not sure whether the assumption is validated.


**Time Spent Reviewing:**

4

---

> ### Author Response · Authors · 2021-08-08
> **Response to Reviewer PdUi**
>
> Thanks for your valuable comments on the paper.
>
> **Q1: "The experimental evaluation could be done with more complex datasets"**
>
> **A1:** We appreciate your pertinent suggestion. However, we do try to cover the most representative datasets in this paper to highlight our contributions to the offline RL problem. The most commonly used benchmark (D4RL) is already considered in our experiment, which can be referred to [1,2,3,4]. We will definitely consider applying our method to more real-world datasets in future work.
>
> **Q2: "The evaluation of the two policies is based on relaxing the trajectories sampled by a single policy, not sure whether the assumption is validated"**
>
> **A2:** We will add more clarifications to our assumption in the revision. Detailed explanations are listed below:
>
> - First, in real-world applications, a policy is usually fixed after it is deployed to the production system, to guarantee stable performance. It can interact with the environment and collect trajectories which can be added to the offline dataset. Based on the collected data, we can perform offline RL and get a better policy, which can be further deployed to the production system. As a result, we argue that it is reasonable to assume the trajectory is sampled by one single policy.
>
> - Second, even if the trajectory is sampled by more than one policy, we can still assume a stochastic behavior policy which captures the mixed behavior of these policies (i.e., matching the occupancy measure of the mixed data), and learns such a behavior policy from sampled data in the trajectory.
>
> **More**: We have added additional experiment results according to the other reviewers' suggestions, which can be seen in the external link (https://anonymous.4open.science/r/coil-rebuttal-5334/COIL_rebuttal.pdf).
>
> **References:**
>
> [1]: Rahul Kidambi, Aravind Rajeswaran, Praneeth Netrapalli, and Thorsten Joachims. Morel: Model-based offline reinforcement learning. NeurIPS 2020 arXiv:2005.05951, 2020.
>
> [2]: Aviral Kumar, Aurick Zhou, George Tucker, and Sergey Levine. Conservative q-learning for offline reinforcement learning. NeurIPS 2020 arXiv:2006.04779, 2020.
>
> [3]: Xue Bin Peng, Aviral Kumar, Grace Zhang, and Sergey Levine. Advantage-weighted regression: Simple and scalable off-policy reinforcement learning. arXiv preprint arXiv:1910.00177, 2019.
>
> [4]: Tianhe Yu, Garrett Thomas, Lantao Yu, Stefano Ermon, James Zou, Sergey Levine, Chelsea Finn, and Tengyu Ma. Mopo: Model-based offline policy optimization. NeurIPS 2020 arXiv:2005.13239, 2020

---

> > ### Comment · Reviewer_PdUi · 2021-08-14
> > **Update**
> >
> > - Thanks for the explanation of the dataset. I agree that the dataset used in the paper are from the existing benchmarks. However, I am not so convinced whether really fits the assumption in this paper.
> >
> > - Thanks for the explanation of the assumption. I think it is make sense.
> >
> > In general, I am happy with authors response, and will be happy to raise my ratings.

---

### Decision · Program_Chairs · 2021-09-27

**Decision:**

Accept (Poster)

**Comment:**

This paper generated an interesting discussion between the reviewers and the authors. Especially, the authors made a very good job at addressing reviewers' concerns. They ran new experiments, finetuned the baselines and conducted a thorough ablation study. The paper is now considered as presenting a SotA algorithm for Offline imitation learning.